# Illumina sequencing of 16S rRNA genes reveals a unique microbial community in three anaerobic sludge digesters of Dubai

**Munawwar Ali Khan**[1◉], **Shams Tabrez Khan**[2◉]*, **Milred Cedric Sequeira**[3], **Sultan Mohammad Faheem**[3], **Naushad Rais**[3]

**1** Department of Life and Environmental Sciences, College of Natural and Health Sciences, Zayed University, Dubai, United Arab Emirates, **2** Department of Agricultural Microbiology, Aligarh Muslim University, Aligarh, Uttar Pradesh, India, **3** School of Life Sciences, Manipal Academy of Higher Education, Academic City, Dubai, United Arab Emirates

◉ These authors contributed equally to this work.
* shamsalig75@gmail.com

**Data Availability Statement:** The raw sequences are available in the National Center for Biotechnology Information (Accession no. SRX10004771-SRX10004780) under the Bio-

## Abstract

Understanding the microbial communities in anaerobic digesters, especially bacteria and archaea, is key to its better operation and regulation. Microbial communities in the anaerobic digesters of the Gulf region where climatic conditions and other factors may impact the incoming feed are not documented. Therefore, Archaeal and Bacterial communities of three full-scale anaerobic digesters, namely AD1, AD3, and AD5 of the Jebel Ali Sewage water Treatment Plant (JASTP) were analyzed by Illumina sequencing of 16S rRNA genes. Among bacteria, the most abundant genus was fermentative bacteria *Acetobacteroides* (Blvii28). Other predominant bacterial genera in the digesters included thermophilic bacteria (*Fervidobacterium* and *Coprothermobacter*) and halophilic bacteria like *Haloterrigena* and *Sediminibacter*. This can be correlated with the climatic condition in Dubai, where the bacteria in the incoming feed may be thermophilic or halophilic as much of the water used in the country is desalinated seawater. The predominant Archaea include mainly the members of the phyla Euryarchaeota and Crenarchaeota belonging to the genus *Methanocorpusculum*, *Metallosphaera*, *Methanocella*, and *Methanococcus*. The highest population of *Methanocorpusculum* (more than 50% of total Archaea), and other hydrogenotrophic archaea, is in agreement with the high population of bacterial genera *Acetobacteroides* (Blvii28) and *Fervidobacterium*, capable of fermenting organic substrates into acetate and $H_2$. *Coprothermobacter*, which is known to improve protein degradation by establishing syntrophy with hydrogenotrophic archaea, is also one of the digesters' dominant genera. The results suggest that the microbial community in three full-scale anaerobic digesters is different. To best of our knowledge this is the first detailed report from the UAE.

Project ID: PRJNA602372. The corresponding Biosample accession numbers are SAMN12749038, SAMN12749039, SAMN12749048, SAMN12749049, SAMN12749050, SAMN12749051, SAMN12749052, SAMN12749053; SAMN12749054; SAMN12749055. The sequence data records can be accessed from the following link: https://www.ncbi.nlm.nih.gov/sra/PRJNA602372.

**Funding:** This work was supported by Zayed University Microbial Diversity and Ecogenomics Research Cluster Project (Grant code: R16092). The funders had no role in study design, data collection and analysis, decision to publish, or preparation of the manuscript.

**Competing interests:** The authors have declared that no competing interests exist.

## Introduction

The process of anaerobic digestion is a multistep microbial process that includes the microbes mediated breakdown of organic matter to produce $CO_2$, $CH_4$, and $H_2O$ by a complex microbial community including archaea and bacteria. These products can be used as biogas, which can be further processed to generate electricity or fuel for transportation [1, 2]. The energy obtained from anaerobic digesters can also be used for the operation of the wastewater treatment plant itself [3]. Due to these possible applications of the gases and the rising cost of conventional fossil fuels, the anaerobic digestion process is emerging as one of the most sustainable methods for the management of organic waste [4]. The process is already being used for the generation of renewable energy in many countries worldwide [5]. Besides, anaerobic digestion is of environmental significance as it helps manage waste and reduce greenhouse gases [6]. There are, however, clear challenges in translating the technology for simple commercial applications and general use [7]. One of the reasons is the complexity of the microbial community involved in the process, making it difficult to understand the specific roles of different bacteria and their maintenance as consortia [8, 9]. Therefore, it is important to understand the roles of various microorganisms in the process of anaerobic digestion.

During the conversion of organic wastes into $CH_4$ gas, a number of microbial processes are involved [10]. The anaerobic digestion process involves the following basic steps: the breakdown of complex organic compounds to simple organic molecules, the conversion of these simple organic molecules to organic acids (acidification), and the conversion of these organic acids into $CH_4$ gas. Methanogens are also of two major types: hydrogenotrophic and acetolactic, depending on the type of substrate they utilize for $CH_4$ production. Each of these processes is carried out by different microorganisms in the presence of various other unrelated microorganisms. The complex interplay between these microorganisms may influence these processes adversely or favorably. Multiple factors influence the composition of the microbial community present in the anaerobic digesters [11]. One important factor can be the microbial community in the feed of the anaerobic digester, which influences the anaerobic digester's microbial community [12, 13]. The composition of the microbial community in the feed can also change with environmental conditions. It has been demonstrated earlier that environmental conditions affect the type of methanogens present in anaerobic digesters [14].

Although several studies have reported the microbial community associated with full-scale anaerobic sludge systems worldwide, exhaustive research based on next-generation sequencing describing the microbial community structure in full-scale anaerobic sludge digesters of the UAE and other wastewater treatment plants (WWTPs) in the Gulf is missing. It is also essential to understand the microbial community in these WWTPs due to two main reasons. First, the climatic conditions in these countries are different, and secondly, much of the water used is desalinated water from the sea, which may influence the microbial community in the incoming feed, consequently affecting the process of digestion. Knowledge of the microbial community in these digesters and its role in the anaerobic digestion process can help in improving the performance of WWTPs, not only in Dubai but also in the whole Gulf region, sharing similar climatic conditions and water resources. Since JASTP is one of the two main municipal sewage treatment plants operating in Dubai, its successful operation is of critical importance. Both treated wastewater and digested sludge produce are reused for various purposes. The biogas produced by anaerobic digesters is partly utilized to maintain digester temperature and run the boiler systems. Dubai also has future plans to use biogas produced from these WWTPs. Therefore, we have previously reported the microbial community in these digesters using the fluorescent *in situ* hybridization and quantitative PCR technique [15]. Here we report a comprehensive mapping of key microbial operational taxonomic units present in these digesters.

To the best of our knowledge, the microbial community of these anaerobic digesters is not documented, especially using next-generation sequencing approaches like Illumina sequencing. Therefore, this study reports a preliminary analysis of the microbial community present in the anaerobic digesters of JASTP, including bacteria and archaea, using Illumina sequencing.

## Materials and methods

### Sampling

Ten waste sludge samples were collected from three anaerobic digesters (AD1, AD3 & AD5) of the Jebel Ali Sewage Water Treatment Plant (JASTP) over a period from September 2016 to February 2017. JASTP is one of the two municipal wastewater treatment plants in Dubai, which serves a population of approximately 3.37 million and processes 375,000 $m^3$ of wastewater per day. The three AD systems chosen for this study are operating for a long time and are considered representative digesters of JASTP and Dubai. Furthermore, the sludge produced during anaerobic digestion is used as a biofertilizer in public parks and local agricultural farms in Dubai. All three digesters had a capacity of 7433 $m^3$ and operated at a mesophilic temperature ranging between 32–37°C [15]. The digesters were fed with 60 and 40% of raw and activated sludge, respectively. Details on the configuration and characteristics of the digesters are listed in Table 1.

The samples used for extraction of genomic DNA for Illumina sequencing were collected simultaneously with other samples used in our previously published studies involving fluorescence *in situ* hybridization (FISH) and real-time PCR assay. The samples collected from three anaerobic digesters were designated as AD1, AD3, and AD5. Temperature, pH, and electrical conductivity (EC) were measured in sludge samples at the time of collection using HORIBA U-50 Multi Water Quality Checker (HORIBA Instruments Incorporated, USA). The collected samples were stored at 4°C until DNA extraction.

### Genomic DNA extraction

Genomic DNA was extracted in triplicate from the sludge samples using the Power Soil DNA Extraction Kit (MO Bio Laboratories, Inc., Solana Beach, CA). The composite samples were

**Table 1. Characteristics and configuration of anaerobic digestors.**

| Characteristics of digester | Digester number | | |
|---|---|---|---|
| | AD1 | AD3 | AD5 |
| Type of digester | CSTR[a] | CSTR | CSTR |
| Operating temperature | mesophilic | mesophilic | mesophilic |
| Input feed | RS[b]-60% | RS-60% | RS-60% |
| | AS[c]-40% | AS-40% | AS-40% |
| Digester capacity ($m^3$) | 7433 | 7433 | 7433 |
| Digester feeding rate ($m^3$/day) | 2248 | 2148 | 2552 |
| Solid retention time (Days) | 16 | 16 | 16 |
| Hydraulic Retention time (Days) | 3.3 | 3 | 2.91 |
| Upflow Velocity ($m^3$/hr) | 120 | 120 | 120 |
| Organic loading rate (Kg. ODS/$m^3$. d) | 6.84 | 5.84 | 6.61 |

[a]CSTR = continuously stirred tank reactors

[b]RS = raw sludge

[c]AS = activated sludge

vortexed and then centrifuged at 4000 rpm for 5 min. DNA was then extracted from 0.25g of pellet according to the manufacturer's protocol. The extracted DNA was stored at -20˚C until further use. DNA concentration and purity were checked using a Qubit fluorometer (Thermo Fisher Scientific, USA).

## Illumina sequencing of samples

The diversity of bacterial and archaeal communities in the samples was determined by amplifying the V3-V4 regions of bacterial and archaeal 16S ribosomal RNA (rRNA) genes. Briefly, quality check of the extracted genomic DNA samples was performed by quantification using the Qubit DNA BR Assay kit (Thermo Fisher Scientific, USA Cat#Q32853). For the generation of 16S amplicon, the extracted DNA samples were diluted to 10 ng and were amplified for 16S (~1500 bp) using 16S (5′ AGAGTTTGATCCTGGCTCAG 3′), & 16S reverse primers (5′ GGTTACCTTGTTACGACTT 3′), positive control (internal metagenomic DNA sample), and no template control. These amplicons were checked on 1% agarose gel. To generate V3-V4 amplicon, 16S amplicon was used as a template with all the samples subjected to V3-V4 amplification (~460 bp) using V3-V4 forward and V3-V4 reverse primers (primer sequences V3-V4-forward 5′ CCTACGGGNGGCWGCAG 3′ and V3-V4 reverse 5′ GACTACHVGGGT ATCTAATCC 3′) and a positive control (internal metagenomic DNA sample), without template control [16]. The amplicons were checked on 1% agarose gel. The V3-V4 amplicons were then cleaned using AMPure XP beads (Beckman Coulter, CA, USA, Cat# A63882) to get rid of non-specific fragments. The V3-V4 products were used for DNA library preparation using NEBNext Ultra DNA Library Prep Kit for Illumina (New England Biolab, UK, Cat# E7370L). First, the amplicons were end-repaired and mono-adenylated at the 3' end in a single enzymatic reaction. Next, NEB hairpin-loop adapters are ligated to the DNA fragments in a T4-DNA ligase-based reaction. Following ligation, the loop containing Uracil is linearized using USER Enzyme (a combination of UDG and Endo VIII), to make it available as a substrate for PCR-based indexing in the next step. During PCR, barcodes were incorporated using unique primers for each of the samples, thereby enabling multiplexing. The prepared libraries were checked for fragment distribution using D1000 Screen Tapes (Cat# 5067–5582, Agilent, CA, USA) and reagents (Cat# 5067–5583, Agilent, CA, USA). The obtained libraries were pooled and diluted to the final optimal loading concentration before cluster amplification on the Illumina flow cell. Once the cluster generation is completed, the clustered flow cell is loaded on Illumina HiSeq2500 instrument (Illumina, Inc., San Diego, USA) for amplicon sequencing to generate 0.5M, 250 bp paired-end reads per sample using the pair-end approach.

## Bioinformatics and statistical analyses

The bioinformatics analysis was carried out using standard methods. Briefly, the following steps were involved. Quality checking of the raw fastq files was carried out using FASTQC to check for the base quality, base composition, and GC content. The sequence reads were trimmed using fastq-mcf to retain only high-quality sequences for further analysis, and the low-quality sequence reads were excluded from the analysis. Sequences were assembled using forward and reverse sequences of the V3-V4 region. Spacer and conserved regions were removed from paired-end reads. Dereplication and the identification of the sequences were carried out using USEARCH. The UCHIME utility from USEARCH was used to remove chimeras using the de novo approach [17]. One representative sequence from each OTU was picked for taxonomic classification using the RDP classifier against the green gene database at 97% similarity. OTUs thus determined were aggregated at the genus level, and all the

downstream processing was carried out at the genus level until unless otherwise mentioned. The raw sequences have been deposited in the National Center for Biotechnology Information (NCBI Bethesda MD, 20894 USA) under Bio-Project accession number PRJNA602372. Alpha diversity in the samples was calculated using the online Calypso program. Two diversity indices, namely Shannon and Chao1 diversity indices, were used to indicate alpha diversity. Venn diagrams showing the genera shared by different samples based on the tables of shared OTUs were also prepared using Calypso. Heat maps were also plotted using calypso, as described earlier [18]. SPSS software version 26 (IBM Corp. Chicago, IL, USA) was used to determine correlations between physicochemical parameters and the predominant bacterial and archaea genera present in the sludges using bivariate Pearson's coefficient of correlation with a level of significance 0.05.

## Results and discussion

### Physicochemical conditions of anaerobic digesters

The physicochemical conditions of the digester operation are given in Table 2.

All three digesters were operated in almost similar physicochemical conditions of mesophilic temperature (34˚C) and a neutral pH range of 7.13–7.55. Among the three digesters, however, AD3 had a relatively low organic loading rate of 5.84 kg. ODS/$m^3$.d compared to AD1 (6.84 kg. ODS/$m^3$. d) and AD5 (6.61 kg. ODS/$m^3$. d). The electrical conductivity (EC) of all sludge digestors was found to be towards the higher range. The highest conductivity of sludge samples was found for AD5 (12.33-15.46 mS $cm^{-1}$) followed by AD3 (11.71-14.44 mS $cm^{-1}$) and AD1 (9.77–13.45 mS $cm^{-1}$). The EC value range found in this study was similar to that observed in another study in Austria evaluating the effects of various co-substrates on the microbial community composition of full-scale anaerobic digesters fed with or without co-substrates [19]. Overall, all three anaerobic digesters showed acceptable levels of the main operational parameters and were performing stably during the sampling period.

### Bacterial community in the anaerobic digesters

The reads per sample varied between 1 million and 1.6 million. After quality filtering, 60% of the total reads were removed, resulting in an average of 0.6 million reads per sample. The sequences were assigned to OTUs based on 97% sequence similarities. The detected OTUs were assigned to 33 phyla, 64 classes, 99 orders, 116 families, and 107 genera. *Bacteroidetes*, *Firmicutes*, *Synergistetes*, *Theromotogae*, OP8, and *Chloroflexi* were dominant in all the digesters, accounting for 90.85% of all sequences (Fig 1).

*Bacteroidetes* were found to be the most abundant phylum (41.44%), followed by *Firmicutes* (23.42%), *Synergistetes* (7.22%), *Thermotogae* (6.70%), *Chloroflexi* (6.69%), and OP8 (5.3%).

Table 2. [a]Physicochemical parameters of anaerobic digesters samples.

| Digester | Temperature (˚C) | pH | EC (mS $cm^{-1}$) | Dry solids (% [wt/vol]) | Volatile solids (% [wt/ wt] of TS) | Volatile fatty acids (mg HAc/L) | Dissolved sulfide (mg/L) | Alkalinity (mg/ L) |
|---|---|---|---|---|---|---|---|---|
| AD1 | 33–34.3 | 7.13–7.33 | 9.77–13.45 | 2.91–3.34 | 70.27–70.95 | 177–195 | 30.6–38 | 3014–3249 |
| AD3 | 34.2–34.5 | 7.22–7.55 | 11.71–14.44 | 2.56–5.74 | 43.75–70.15 | 153–205 | 32.40–37.20 | 2992–3512 |
| AD5 | 32.9–34.4 | 7.36–7.50 | 12.33–15.46 | 2.79–3.54 | 50.3–67.49 | 145.5–195 | 16.80–47.20 | 2893–3498 |

[a]minimum-maximum range

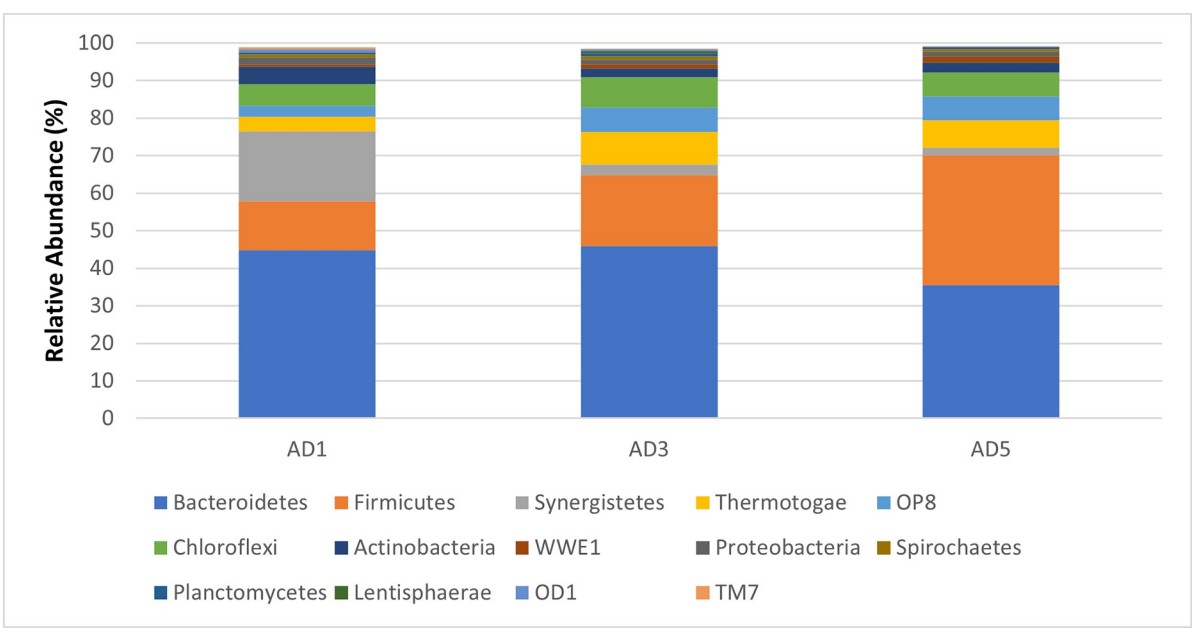

**Fig 1. The relative abundance of different bacterial phyla in the three anaerobic digester samples.**

Other phyla, with a population of >1% were *Actinobacteria* (3.1%), *Proteobacteria* (1.4%), and WWE1 (1.21%). The overall abundance of bacterial community phyla observed is similar to those from anaerobic sludge digester samples in previous studies [19, 20]. However, unlike these studies, the percentages of *Bacteroidetes* (41.44%) detected in this study were significantly high, and the population of proteobacteria was very low (1.4%). The higher population of *Bacteroidetes* phyla can be associated with high hydrolytic activity in full-scale anaerobic digesters [21]. Degradation of macromolecules is the first step in the digestion process, and a high population of Bacteroidetes indicates effective degradation (Fig 2).

A study reported the effect of high salinity on reducing the abundance of members of *Bacteroidetes* and *Chlorflexi* [22]. However, in our study, despite high electrical conductivity, indicative of higher salt concentration observed in three digesters, the population of *Bacteroidetes* remained high i.e., in the range of 44.86% (AD1), 45.95% (AD3), and 35.49% (AD5). The *Synergistetes* phylum was found to be abundant in AD1 (18.67%) compared to AD 3 (2.76%) and AD5(1.98%). The members of *Synergistetes* phyla are known to use amino acids for producing short-chain fatty acids and sulphate to methanogens and sulphate-reducing bacteria [23]. The members of the fourth most abundant phyla *Thermotogae* were previously reported to be linked to polysaccharide fermentation and hydrogen production, which may promote the population of hydrogenotrophic methanogens, as shown in Fig 2 [24]. A previous study has also reported the syntrophic association between methanogenic archaea and the members of *Thermotogae* [25].

A total of 108 bacterial genera were detected in all three samples studied. In addition to these genera, many of the bacteria were grouped as unknown bacteria. Alpha diversity calculated based on the populations of various genera in these samples in terms of Shannon (p = 0.72) and Chao1 (p = 0.48) indices are shown in Fig 3A and 3B, respectively.

The figures show that the lowest diversity is observed in the AD1 sample, followed by AD3 and AD5. Members of 51 bacterial genera were found to constitute the core microbiome of the three digesters studied (Fig 4A). Some of the predominant genera include *Acetobacteroides*

| Major taxa | AD1 | AD3 | AD5 | Process/products |
|---|---|---|---|---|
| Bacteroidetes | 44.86 | 45.95 | 35.49 | Macromolecule degradation |
| Firmicutes | 12.97 | 18.88 | 34.69 | |
| Synergistetes | 18.67 | 2.77 | 1.98 | |
| Thermotogae | 3.87 | 8.69 | 7.34 | |
| Proteobacteria | 1.87 | 1.24 | 1.31 | VFA & Alcohols |
| Acetobacteroides (Blvii28) | 15.83 | 15.35 | 14.68 | |
| Fervidobacterium | 3.27 | 7.74 | 7.20 | |
| Clostridium | 0.45 | 0.40 | 0.20 | |
| Syntrophomonas | 0.03 | 0.02 | 0.03 | |
| Paludibacter | 0.27 | 0.44 | 0.17 | |
| Mycobacterium | 0.61 | 0.19 | 0.19 | Acetate, $H_2$ & $CO_2$ |
| Methanocorpusculum | 0.80 | 0.85 | 0.86 | |
| Metallosphaera | 0.08 | 0.07 | 0.05 | |
| Methanocella | 0.02 | 0.01 | 0.02 | |
| Methanococcus | 0.01 | 0.01 | 0.01 | Methane |

**Fig 2. Abundant microbial genera and their possible involvement in the process of anaerobic digestion.**

(Blvii28), *Coprothermobacter*, *Fervidobacterium*, *Clostridium*, *Caldilinea*, *Allochromatium*, *Sediminibacter*, and T78 (Fig 4B).

Most of these genera are associated with an anaerobic digester; some have also been reported from anaerobic digester earlier also [26]. *Acetobacteroides* (Blvii28) was the genus with the highest population. The cultured representative members of the genus are known to produce acetate, $H_2$, and $CO_2$ as fermentation end-products [27]. As shown in Fig 2, this bacterium may be involved in the production of acetate, $H_2$ and $CO_2$, playing an important role in digestion and promoting the growth of hydrogenotrophic methanogens. T78 possibly metabolizes carbohydrates and alcohol via syntrophic interactions [28]. Many of the predominant bacteria found in this study were thermophilic or halophilic. The presence and dominance of diverse halotolerant bacteria with hydrolytic and acidogenic abilities adapted to the high salt concentrations has been reported earlier [22].

*Coprothermobacter*, a known proteolytic anaerobic thermophilic bacteria, is found in many thermophilic anaerobic digesters [29]. This genus can improve protein degradation by establishing syntrophy with hydrogenotrophic archaea [29]. Other thermophilic bacteria found, include *Fervidobacterium* and *Caldilinea* that are known to ferment carbohydrates to lactate, acetate, hydrogen, and carbon dioxide, as shown in Fig 2 [30, 31]. The genus *Sediminibacter*, initially isolated from a sediment sample, was also found as one of the dominant genera [32]. Since most of the water used in Dubai is obtained from the sea after desalination, the incoming feed may contain such bacteria [33, 34].

Interestingly, *Sediminibacter* has a unique light-driven sodium ion pump that helps in its survival in the marine habitat [35]. Several purple sulfur bacteria like *Allochormatium* and *Thermotogales* AUTHM297 were also part of the digester microbial communities. *Desulfomicrobium* and *Desulfobacter* were the two most dominant sulfate-reducing bacterial (SRB) genera. In our previous study, we have reported consistently high populations of these two genera in the same digesters as detected by fluorescent *in-situ* hybridization [36]. Notably, *Desulfomicrobium* is also known to be associated with the marine habitat [37]. The statistical correlation analysis between physicochemical parameters and bacterial genera indicated that the

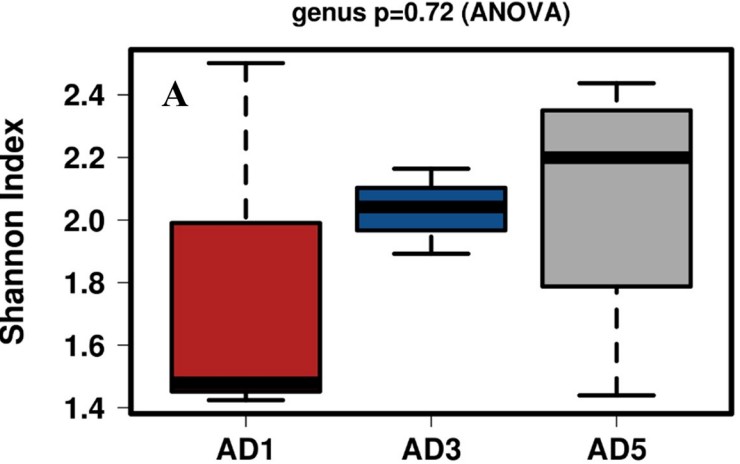

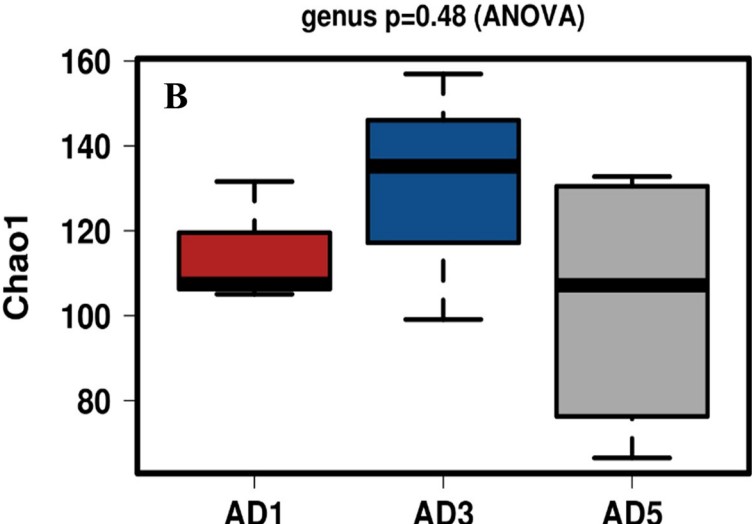

**Fig 3.** Boxplot of Shannon index (A) and Chao 1 (B) showing Alpha diversity in the three anaerobic digesters (AD1, AD3, and AD5) calculated based on the populations of different bacterial genera.

population of *Planctomyces* was significantly correlated with dry solids. In contrast, the W5 genus was associated with the anaerobic digester's operating pH (S1 Table).

Anaerobic digestion involves the degradation of complex organic matter to simple organic compounds, most likely carried out by the members of the phyla *Bacteroidetes*, *Firmicutes*, *Proteobacteria*, *Synergistetes*, and *Thermotogae*. Previous studies also demonstrate the presence of these phyla in anaerobic digesters as macromolecules-degrading bacteria [26]. *Mycobacterium*, which is associated with the production of Lipases and Lipolytic activity, producing fatty acids was also present in high numbers. Another important process of anaerobic digestion is acidogenesis, wherein bacteria convert organic monomers into acids like acetic, propionic, and butyric acids (Fig 2).

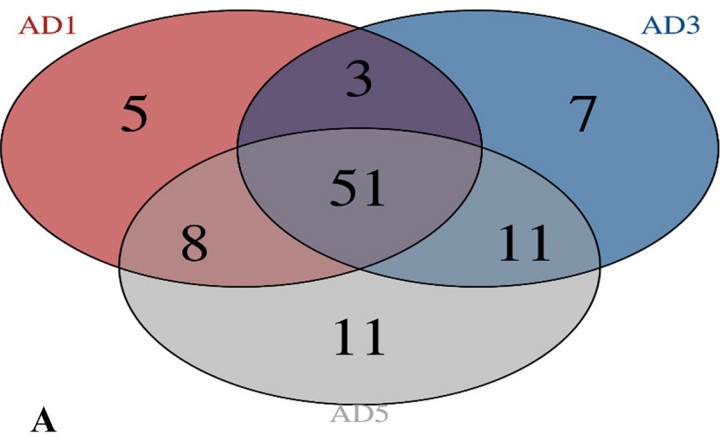

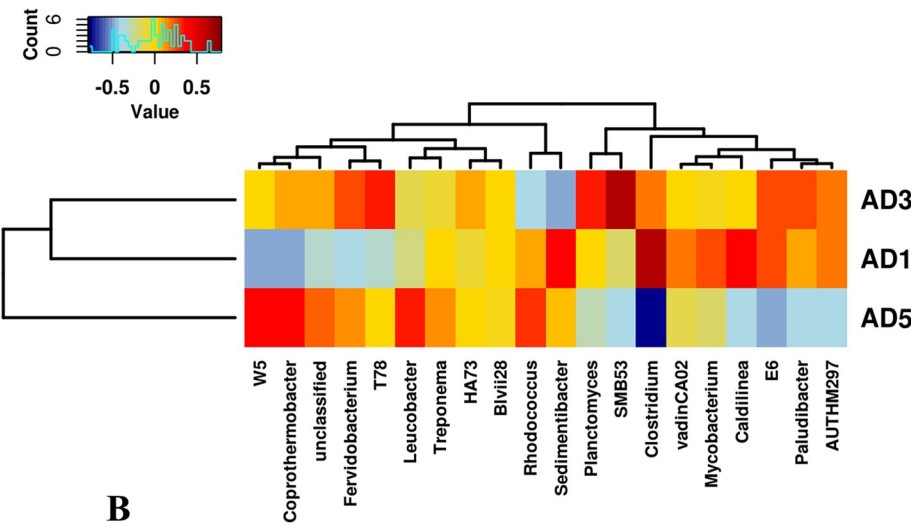

**Fig 4.** Panel A: Venn diagram showing the core microbiome and number of genera shared by the samples. Panel B: shows heat map of hierarchical clustering of twenty genera with the highest mean relative abundance across the three anaerobic digesters.

The populations of acidogenic bacteria like *Acetobacteroides*, *Fervidobacterium*, *Clostridium*, and *Paludibacter* producing acetate, lactate, or propionate were high in the digesters as reported above. The conversion of these organic molecules to methane ($CH_4$) is carried out mainly by archaea. However, some bacteria may influence the production of $CH_4$ by competing with the acetoclastic methanogens. An example of such acetate-utilizing uncultured bacteria is the *Synergistes* group 4 [38].

## Archaea in the anaerobic digesters

When analysed at the genus level, the highest diversity was observed in digester AD5, followed by AD1 and AD3. This was also evident from Shannon index's, and Chao 1 values obtained for AD1 followed by AD3 and AD5 (Fig 5A and 5B). The boxplot of Shannon (A) and Chao 1

**A**

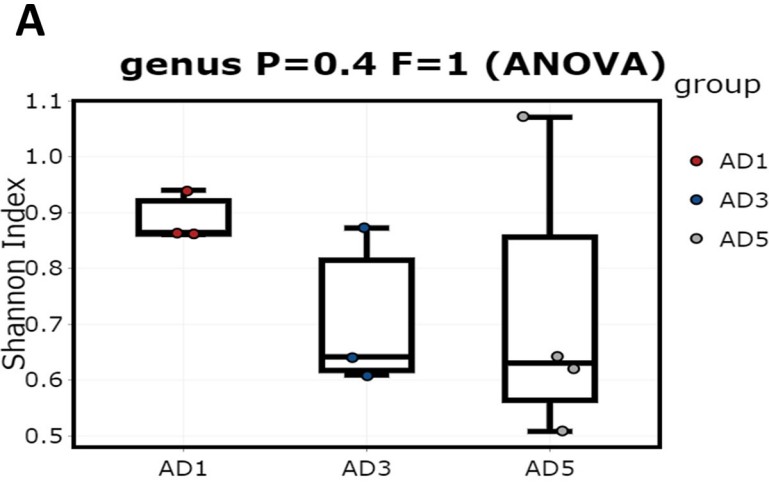

**B**

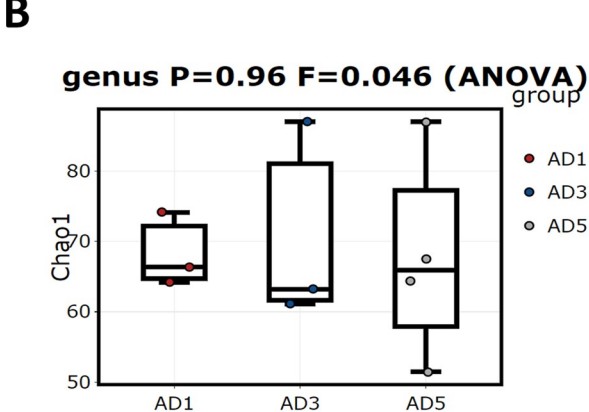

**Fig 5.** The alpha diversity of archaea, Shannon (A), and Chao 1 (B) indices.

(B) indices was calculated based on the populations of different archaeal genera (AD1, AD3, and AD5).

The predominant archaea in the three digesters were the members of the phylum *Euryarchaeota*, followed by *Crenarchaeota*. The three digesters share a core archaeal microbiome of 18 genera (Fig 6A). The population of hydrogenotrophic archaea was clearly high in the digesters. *Methanocorpusculum* alone constitutes more than 50% of the total population. The predominant core genera of archaea included *Metallosphaera, Methanocella, Methanococus, Acidianus, Natronobacterium*, and others. *Metallosphaera* was significantly correlated with the dry solids parameter of the anaerobic digesters (S2 Table).

Fig 6B shows the heat map of predominant archaea in the three digesters. *Methanocorpusculum* is one of the hydrogenotrophic methanogens which was isolated for the first time from the biodigester of the wastewater treatment plant [39]. This archaea genus and other

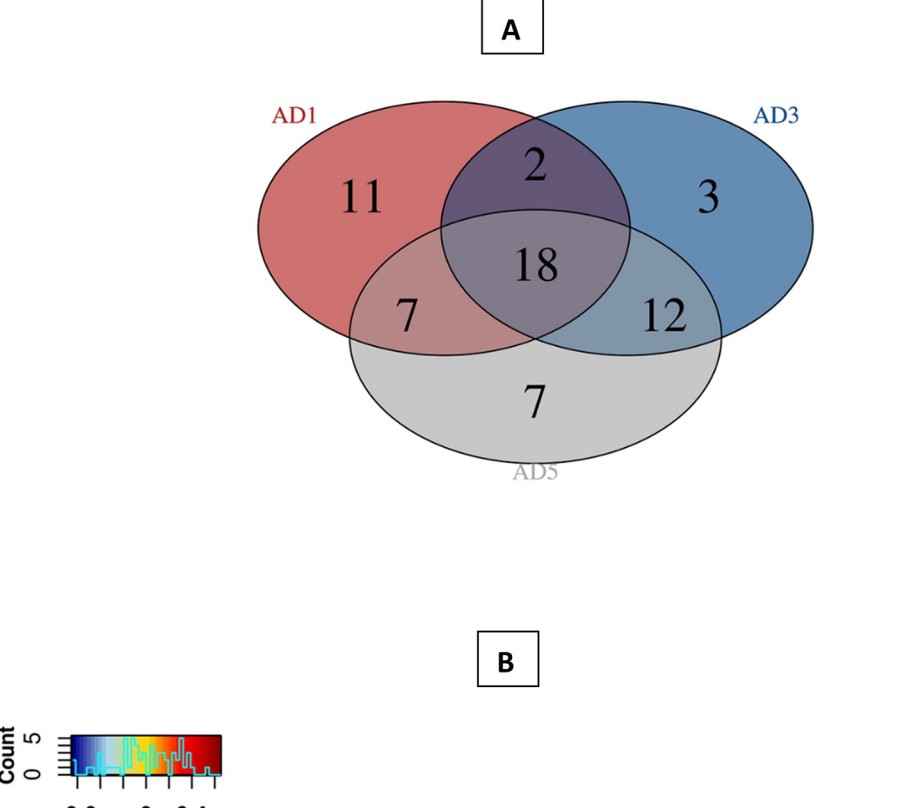

**Fig 6.** Panel A: Venn diagram showing the core Archaeal genera present in the three anaerobic digesters (AD1, AD3, and AD5) and the number of genera shared by different digesters. Panel B: Heat map of hierarchical clustering of twenty archaeal genera with the highest mean relative abundance across the three anaerobic digesters.

hydrogenotrophic methane-producing archaea may utilize the $H_2$ produced by a predominant bacterial genus (*Acetobacteroides*; Blvii28) were present in the digesters (Fig 2). Interestingly, the population of *Acetobacteroides* (Blvii28) was significantly related to the population of many other predominant hydrogenotrophic archaea in the digesters, including *Methanococcus*

and *Methanothermus* (S3 Table). The analysis shows that the population of *Methanocorpusculum* was also significantly correlated with the population of *Coprothermobacter*, which is known to improve protein degradation by establishing a syntrophic relationship with the hydrogenotrophic archaea (S3 Table).

While the second most predominant genus was *Metallosphaera*, an extreme thermoacidophile with optimal growth at 74˚C and pH 2.0 [40], *Methanocella* was the third most dominant genus and is a mesophilic, hydrogenotrophic methanogen [41]. At the same time, *Methanococcus* is also a thermophilic hydrogenotrophic methanogen. At least one species of the genus is also known to fix nitrogen [38]. The *Methanothermus fervidus* species found abundantly are hyperthermophiles methanogen, reported for thermophilic anaerobic digestion of wastewater sludge [42]. Several non-methanogenic archaeal genera members were found in very high populations in our study. Notable among these were genera *Acidianus*, an archaeal genus found to inhabit halophilic, hyperthermophilic, and acidic environments. One of the unique genera of extremophile detected in our study was *Natronobacterium*, a haloalkaliphilic archaeon found in extremely hypersaline lakes [43]. *Natronobacterium gregoryi* species found in this study have not been reported from anaerobic digesters operating at mesophilic temperatures and neutral pH. The genus *Natronobacterium* are facultative anaerobes with the capability to reduce sulfur under extremely halophilic and alkaline conditions [44]. The archaeal genus *Ferroplasma* is known to grow at highly acidic pH levels and tolerate heavy metals such as Cu. As, Cd, Zn. This was another genus of extremophilic archaea that showed predominance in our study [45].

The studied digesters contained high populations of the *Crenarchaeota* genera *Metallosphaera*, *Acidianus*, and *Sulfolobus*. These genera contain thermophilic enzymes involved in the conversion of $CO_2$ into liquid fuels and industrial chemicals and hence can be engineered for the production of value-added compounds like 3-hydroxypropionate or n-butanol [46]. *Candidatus Nitrosocaldus*, a novel ammonia-oxidizing archaea genus, was among the top 14 dominating genera found in this study. The genus is a member of the phylum *Thaumarchaeota* and is generally found in geothermal environments [47]. However, recently its presence in the anaerobic digester has been reported [48].

It is clear from the analysis that a unique and atypical microbial community exists in the anaerobic digesters studied. The degradation is carried out by the members of various phylum reported earlier. These include the members of the phyla *Bacteroidetes*, *Firmicutes*, *Proteobacteria*, *Synergistetes*, and *Thermotogae* [13]. In contrast, a number of genera mainly involved in the conversion of these organic substrates into acetate, butyric acid, propionate, and $H_2$ were found in the anaerobic digesters. *Euryarchaeota*, mainly the hydrogenotrophic methanogens, were found as predominant members (Fig 6). Although the digesters were operated under mesophilic conditions, many thermophilic genera of bacteria and archaea were found to predominate.

Furthermore, high populations of various halophilic and acidophilic bacteria and archaea were also detected. This may be due to the presence of such bacteria in the incoming feed, as most of the water used in Dubai is obtained from the sea and is used after desalination. Whether such bacteria are initially present in the incoming feed is a matter of future investigation. Similar observations were made in an earlier study that demonstrated how the incoming feed shapes the microbial community structure in anaerobic digesters [49], through the use of different microbial inoculants. Our study here argues that the microbial community present in the incoming feed is influenced by local environmental conditions, which consequently play important roles in shaping the microbial community in anaerobic digesters.

## Conclusions

The bacterial and archaeal community structure of three full-scale anaerobic sludge digesters of a municipal sewage treatment plant in Dubai, UAE, were compared using Illumina sequencing of 16S rRNA genes. *Bacteroidetes*, *Firmicutes*, *Synergistetes*, *Theromotogate*, OP8, and *Chloroflexi* were dominant bacterial phyla in all the digesters. The highest diversity was observed in AD5, followed by AD3 and AD1. The predominant archaea included mainly the members of phyla *Euryarchaeota* and *Crenarchaeota*. Members of the genus *Methanocorpusculum*, *Metallosphaera*, *Methanocella*, and *Methanococcus* were predominant. The highest diversity of archaea was also observed in AD5, followed by AD3 and AD1. The presence of many thermotolerant and halotolerant bacteria and archaea in the anaerobic digesters may be due to the influence of environmental conditions on the incoming feed sludge. The anaerobic digesters were also characterized by a high population of bacteria known to ferment organic substrates to acetate and $H_2$. The high population of hydrogenotrophic archaea can utilize the $H_2$ produced by these bacteria. Comparison of microbial communities in these three digesters shows that they contain a core, stable, and functional microbial community as all digesters were operated under more or less similar physicochemical conditions. Furthermore, understanding the microbial communities in these digesters can help design strategies for the better performance of the digesters in Dubai and neighbouring countries sharing similar climatic conditions and water resources.

## Supporting information

**S1 Table. Correlation between bacterial genera and physicochemical parameters.**
(XLSX)

**S2 Table. Correlation between archaeal genera and physicochemical parameters.**
(XLSX)

**S3 Table. Correlation between archaeal and bacterial genera.**
(XLSX)

## Acknowledgments

The authors wish to thank Jebel Ali Sewage Treatment Plant authority for providing samples and operational parameters data of anaerobic digesters used in this study. We are thankful for feedback from James Peter Terry for improving the manuscript language.

## Author Contributions

**Conceptualization:** Munawwar Ali Khan, Shams Tabrez Khan.

**Formal analysis:** Munawwar Ali Khan, Shams Tabrez Khan, Milred Cedric Sequeira, Naushad Rais.

**Investigation:** Milred Cedric Sequeira.

**Methodology:** Munawwar Ali Khan, Milred Cedric Sequeira.

**Resources:** Sultan Mohammad Faheem.

**Software:** Naushad Rais.

**Supervision:** Munawwar Ali Khan, Sultan Mohammad Faheem.

**Writing – original draft:** Munawwar Ali Khan, Shams Tabrez Khan.

**Writing – review & editing:** Munawwar Ali Khan, Shams Tabrez Khan.

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
