## [Decision Letter · Decision Letter 0]

29 Jan 2021

PONE-D-20-37224

Illumina sequencing of 16S rRNA genes reveals a unique microbial community in three anaerobic sludge digesters of Dubai.

PLOS ONE

Dear Dr. Khan,

Thank you for submitting your manuscript to PLOS ONE. After careful consideration, we feel that it has merit but does not fully meet PLOS ONE’s publication criteria as it currently stands. Therefore, we invite you to submit a revised version of the manuscript that addresses the points raised during the review process.

We look forward to receiving your revised manuscript.

Kind regards,

William J Brazelton

Academic Editor

PLOS ONE

Journal Requirements:

4.Thank you for stating the following in the Funding Section of your manuscript:

"This work was supported by Zayed University Microbial Diversity and Ecogenomics Research Cluster Project (Grant code: R16092)."

Additional Editor Comments:

Both reviewers recommend rejection, but one of their key reasons for rejection, "significance", is not a valid reason for rejection from PLoS ONE. Moreover, I think that the reported data could be valuable contributions to the public databases and to the literature if properly discussed in the paper. Therefore, if you can make substantial revisions to the manuscript that address the reviewers' concerns, then I will consider a heavily revised manuscript. The revised manuscript must explain how the current study is related to previous work with the same or similar samples. It must attempt to interpret the results from the perspective of trying to understand the microbial processes in anaerobic digester systems, rather than simply concluding that the microbial communities are "different". All microbial communities are different from each other. All microbial communities are unique. The revised manuscript should explain how the reported microbial diversity data helps to improve our knowledge of anaerobic digesters. Finally, the authors should make their data publicly available. Currently, the provided BioProject accession is not associated with any sequence data that I can see.

The issues listed above are the minimum that should be addressed for a revised manuscript to be re-submitted. If the revision does not address all of the concerns from the reviewers, then it may be rejected even after revision. If you require additional time to make all necessary revisions, an extension can be requested.

Reviewers' comments:

Reviewer's Responses to Questions

**Comments to the Author**

1. Is the manuscript technically sound, and do the data support the conclusions?

Reviewer #1: Yes

Reviewer #2: Partly

2. Has the statistical analysis been performed appropriately and rigorously? 

Reviewer #1: Yes

Reviewer #2: No

3. Have the authors made all data underlying the findings in their manuscript fully available?

Reviewer #1: Yes

Reviewer #2: Yes

4. Is the manuscript presented in an intelligible fashion and written in standard English?

Reviewer #1: Yes

Reviewer #2: Yes

5. Review Comments to the Author

Reviewer #1: This study investigated the microbial community in three anaerobic digesters of JASTP using the Illumina sequencing. This article simply describes the species and abundance of archaea and bacteria in three anaerobic digesters. The content is relatively simple and has strong regional limitations. In addition, this study is not of great significance. Hence, it is not suggested to be published in PONE.

Comments:

1. What is the significance of this study to research the microbial distribution in these three anaerobic digesters? Why did the author choose these three anaerobic digesters of JASTP? Are they representative? What reference can this study provide for sludge disposal in Dubai in the future?

2. “Fig 1” should be corrected as “Fig. 1”, so as the others.

3. there are write mistakes in line 205, 585.

4. there are some format mistakes in the References.

Reviewer #2: The authors investigated the microbial community (Bacteria and Archaea) of three anaerobic digesters treating sewage water in Dubai by illumina sequencing.

In my opinion, in order to be published in PLOS ONE the paper need a deep revision by the authors and, after that, a re-submission.

The biggest issues are related to:

Some general mistakes (eg: line 67 Methanogens (methane-producing bacteria)

The lack of information about replicates, sampling period etc

Some data in the results section that were already present in Khan et al. 2019 (Table 1 and 2). To this regard, I suggest to clarify if the analysed digesters and samples are the same (same extracted DNA?). Accordingly, with this I suggest to report explicitly in the discussion differences or similarities of the two experiments.

Moreover, the discussion need to be deepen. As currently, that is not enough. In any case, the bacterial and archaeal populations results should be considered in relation to digester inlet (wastewater) and outlet (digestate and biogas).

In my opinion, in the current form, the paper is not enough significant for the current state of the art of anaerobic digestion to be published in PLOS ONE

6. PLOS authors have the option to publish the peer review history of their article (what does this mean?). If published, this will include your full peer review and any attached files.

Reviewer #1: No

Reviewer #2: No

---

## [Author Response · Author response to Decision Letter 0]

14 Feb 2021

Response to Reviewer 1 Comments

Comment-1: This study investigated the microbial community in three anaerobic digesters of JASTP using the Illumina sequencing. This article simply describes the species and abundance of archaea and bacteria in three anaerobic digesters. The content is relatively simple and has strong regional limitations. In addition, this study is not of great significance. Hence, it is not suggested to be published in PONE.

Response:

JASTP is one of the two main municipal sewage treatment plants operating in Dubai. The successful operation of JASTP is of critical importance to Dubai since both wastewater and digested sludge is reused for further application. The biogas produced by anaerobic digesters of the JASTP is partly utilized to maintain digester temperature and run the boiler systems. However, there are plans in Dubai for the future to capture and use of biogas produced during the treatment of domestic organic waste to energy. However, understanding microbial community structure and functions of anaerobic digester systems in this part of the world is lacking. The results reported in our manuscript not only describe the microbial community in these digesters but also helps us to predict the roles of various microorganisms detected in the treatment of sludge during the anaerobic digester stage of the wastewater treatment process. This knowledge will be beneficial for devising appropriate control methods, leading to more efficient operation of anaerobic digester systems of wastewater treatment plants in the UAE. Furthermore, as reported in the manuscript, microorganisms involved in the process seem to be influenced by the local environmental condition and the fact that most of the water used in the country is desalinated water from the sea. This finding has much greater implications as in most Gulf countries; the conditions are almost similar. Furthermore, so far, no such study to our knowledge is available on this aspect from the region. In light of the above discussion, the study is not merely an isolated study of some digesters in Dubai, but it will provide a basis for the whole region.

Comment-2: What is the significance of this study to research the microbial distribution in these three anaerobic digesters? Why did the author choose these three anaerobic digesters of JASTP? 

Are they representative? What reference can this study provide for sludge disposal in Dubai in the future?

Response: As discussed above JASTP is one of the two main municipal sewage treatment plants in Dubai. The three AD systems chosen for this study are operating for a long time and are considered representative digesters of JASTP and Dubai. The significance of the study is manifold, as discussed above. It is the first study of this type, which may have greater implications for the ADs of the gulf region. The climatic condition and the type of water used may influence the process of anaerobic digestion. Therefore, this study's findings not only provide a reference to the sludge disposal in Dubai but also to other gulf countries sharing similar climatic condition and water quality. 

Since the sludge produced during anaerobic digestion is used as a biofertilizer in public parks and local agricultural farms in Dubai. Which may further affect the soil microbial communities and soil fertility. Ensuring stable operation of the digesters is of critical importance to the city of Dubai making it necessary to understand the microbial community involved. 

Comment-3: “Fig 1” should be corrected as “Fig. 1”, so as the others.

Response: Suggested style for citing figures for PLOS ONE journal is “Fig 1”. We have followed journal formatting style for the figures.

Comment-4: there are write mistakes in line 205, 585.

Response: We have corrected mistakes in line 205 and 585.

Comment-5: there are some format mistakes in the References.

Response: The reference format is corrected as per the PLOS One formatting style using EndNote software.

Response to Reviewer 2 Comments

Comment-1: The authors investigated the microbial community (Bacteria and Archaea) of three anaerobic digesters treating sewage water in Dubai by illumina sequencing.

In my opinion, in order to be published in PLOS ONE the paper need a deep revision by the authors and, after that, a re-submission.

Response: We have thoroughly revised the manuscript as per the reviewer's suggestion.

Comment-2: The biggest issues are related to: Some general mistakes (eg: line 67 Methanogens (methane-producing bacteria)

Response: This error has been corrected.

Comment-3: The lack of information about replicates, sampling period etc

Response: We have included this missing information in the methodology section between lines 88 to 90.

Comment-4: Some data in the results section that were already present in Khan et al. 2019 (Table 1 and 2). To this regard, I suggest to clarify if the analysed digesters and samples are the same (same extracted DNA?). Accordingly, with this I suggest to report explicitly in the discussion differences or similarities of the two experiments.

Response:

The samples used for extraction of genomic DNA for Illumina sequencing were different but collected simultaneously with samples used in our previous published studies involving Fluorescence in situ hybridization FISH and real-time PCR assay. JASTP laboratory has a weekly or biweekly schedule for analysis of digester operational parameters. JASTP officials provided the reported digesters parameters for that sampling period. As suggested, we have reported it clearly in the introduction section (lines 78-84).

Comment-5: Moreover, the discussion need to be deepen. As currently, that is not enough. In any case, the bacterial and archaeal populations results should be considered in relation to digester inlet (wastewater) and outlet (digestate and biogas).

Response: We have improved the discussion section. The collected digester samples are composite samples. There was only one accessible sampling port used for digester parameters analysis.

Comment-6: In my opinion, in the current form, the paper is not enough significant for the current state of the art of anaerobic digestion to be published in PLOS ONE.

Response:

We have improved the manuscript considerably. We have already discussed the manifold significance of the study. The authors hope that the revised manuscript will be of great interest to its readers and attract readership, especially in the gulf region. This study will also attract readers elsewhere as there is no report to our knowledge on the Gulf region's ADs with particular regard to its climatic condition. 

Response to Additional Editor Comments

Journal Requirements:

Response 1: We have reformatted the manuscript to meet PLOS ONE's style requirements.

2. PLOS requires an ORCID iD for the corresponding author in Editorial Manager on papers submitted after December 6, 2016. Please ensure that you have an ORCID iD and that it is validated in Editorial Manager. To do this, go to ‘Update my Information’ (in the upper left-hand corner of the main menu), and click on the Fetch/Validate link next to the ORCID field. This will take you to the ORCID site and allow you to create a new iD or authenticate a pre-existing iD in Editorial Manager. Please see the following video for instructions on linking an ORCID iD to your Editorial Manager account: https://www.youtube.com/watch?v=_xcclfuvtxQ

Response 2: The corresponding author ORCID iD has been updated in the Editorial Manager.

Response 3: We have added supporting table information on the last page of the manuscript and amended in-text citations of supporting information according to the journal's guidelines.

4.Thank you for stating the following in the Funding Section of your manuscript:

"This work was supported by Zayed University Microbial Diversity and Ecogenomics Research Cluster Project (Grant code: R16092)."

Response 4: This is true that the funders had no role in study design, data collection and analysis, decision to publish, or preparation of the manuscript. However, the work would not have been possible without the generous financial support from the funding agency, that is why we want to acknowledge the funding agency. We have moved funding-related information to the acknowledgement section in the revised manuscript. 

Additional Editor Comments:

Point 1: Both reviewers recommend rejection, but one of their key reasons for rejection, "significance", is not a valid reason for rejection from PLoS ONE. Moreover, I think that the reported data could be valuable contributions to the public databases and to the literature if properly discussed in the paper. Therefore, if you can make substantial revisions to the manuscript that address the reviewers' concerns, then I will consider a heavily revised manuscript. 

Response: We appreciate the opportunity provided by the editor to considerably improve our manuscript for possible publication in PlosOne. Keeping in view the comments from the editor and reviewers the manuscript has been improved significantly. I hope the revised manuscript meets the standard of PlosOne and will be published in the journal. 

Point 2: The revised manuscript must explain how the current study is related to previous work with the same or similar samples. 

It must attempt to interpret the results from the perspective of trying to understand the microbial processes in anaerobic digester systems, rather than simply concluding that the microbial communities are "different". All microbial communities are different from each other. All microbial communities are unique. The revised manuscript should explain how the reported microbial diversity data helps to improve our knowledge of anaerobic digesters. 

Response: The sludge samples for the next-generation sequencing analysis, FISH and qPCR analysis were collected simultaneously from the digesters. This information has been added to the revised manuscript. We agree to the editors comment, and we have elaborated the possible role of various microorganisms in the process of anaerobic digestion as also summarised in figure 4. Further information on the role of these microorganisms in the process of anaerobic digestion is also added to the revised manuscript, especially in the discussion section. We have concluded in the manuscript that the processes are mediated by the halophilic and/or thermophilic microorganisms which may be due to the influence of the local climatic condition. This study, therefore, is a primer for the possible role of microorganisms in the process of anaerobic digestion in gulf region. In future studies, we can try to isolate such halophilic or thermophilic bacteria from local environment that can help to improve the process of anaerobic digestion. In future we will do metagenome based study of the same digesters to understand the functional aspects more clearly.

Point 3: Finally, the authors should make their data publicly available. Currently, the provided BioProject accession is not associated with any sequence data that I can see.

The issues listed above are the minimum that should be addressed for a revised manuscript to be re-submitted. If the revision does not address all of the concerns from the reviewers, then it may be rejected even after revision. If you require additional time to make all necessary revisions, an extension can be requested.

Response: We have updated the deposited Nucleotide sequences files in the National Center for Biotechnology Information (Accession no. SRX10004771-SRX10004780) under the Bio-Project ID: PRJNA602372 on February 3, 2021. The sequence data records are now open to the public and accessible with the following link:

https://www.ncbi.nlm.nih.gov/sra/PRJNA602372

---

## [Editor Report · Decision Letter 1]

22 Feb 2021

PONE-D-20-37224R1

Illumina sequencing of 16S rRNA genes reveals a unique microbial community in three anaerobic sludge digesters of Dubai.

PLOS ONE

Dear Dr. Khan,

Thank you for submitting your manuscript to PLOS ONE. After careful consideration, we feel that it has merit but does not fully meet PLOS ONE’s publication criteria as it currently stands. Therefore, we invite you to submit a revised version of the manuscript that addresses the points raised during the review process.

Thank you for revising the manuscript. The revision addresses some but not all of the reviewer concerns. The responses to the reviewer's comments (e.g. the general importance of this study for effective wastewater management in Dubai and why these digesters were chosen) are not fully reflected in the text. Please ensure that the points made in your responses to the reviewers are also in the manuscript text. Also, Figure 4 contains a nice summary of some key results, but this figure is only briefly cited in the text. Please incorporate this figure more fully into the Results and Discussion. Finally, please check the entire manuscript for English grammar. For example, multiple sentences beginning with "Although" and "But" are not complete sentences.

We look forward to receiving your revised manuscript.

Kind regards,

William J Brazelton

Academic Editor

PLOS ONE

---

## [Author Response · Author response to Decision Letter 1]

28 Feb 2021

Response to Reviewer’s /Additional Editor Comments

Comment: Thank you for submitting your manuscript to PLOS ONE. After careful consideration, we feel that it has merit but does not fully meet PLOS ONE’s publication criteria as it currently stands. Therefore, we invite you to submit a revised version of the manuscript that addresses the points raised during the review process.

Response: Authors thank the reviewers for providing an opportunity to revise the manuscript. The manuscript is thoroughly revised keeping in view the suggestions of reviewers and academics editor. Manuscript is also considerably improved to avoid grammatical, language and editing mistakes.

Comment: Thank you for revising the manuscript. The revision addresses some but not all of the reviewer concerns. The responses to the reviewer's comments (e.g. the general importance of this study for effective wastewater management in Dubai and why these digesters were chosen) are not fully reflected in the text. Please ensure that the points made in your responses to the reviewers are also in the manuscript text.

Response: The manuscript is revised to include the importance of the study in the manuscript text. Furthermore, we have also modified the manuscript to include the changes suggested by reviewers earlier.

Comment: Also, Figure 4 contains a nice summary of some key results, but this figure is only briefly cited in the text. Please incorporate this figure more fully into the Results and Discussion.

Response: A detailed and duly discussion on figure 4 (now figure 2) is now included in the revised manuscript. Figure 4 is renamed as figure 2 to follow the in-text figure sequence style of the journal.

Comment: Finally, please check the entire manuscript for English grammar. For example, multiple sentences beginning with "Although" and "But" are not complete sentences.

Response: The manuscript is thoroughly checked for grammatical and editing mistakes by a native speaker. The authors hope that the language is significantly improved in the revised manuscript.

---

## [Editor Report · Decision Letter 2]

10 Mar 2021

Illumina sequencing of 16S rRNA genes reveals a unique microbial community in three anaerobic sludge digesters of Dubai.

PONE-D-20-37224R2

Dear Dr. Khan,

We’re pleased to inform you that your manuscript has been judged scientifically suitable for publication and will be formally accepted for publication once it meets all outstanding technical requirements.

Kind regards,

William J Brazelton

Academic Editor

PLOS ONE

Additional Editor Comments (optional):

Thank you very much for responding to all of the comments and concerns. I think the manuscript is much improved, and I hope you agree. Best wishes.

---

## [Editor Report · Acceptance letter]

23 Mar 2021

PONE-D-20-37224R2 

Illumina sequencing of 16S rRNA genes reveals a unique microbial community in three anaerobic sludge digesters of Dubai. 

Dear Dr. Khan:

I'm pleased to inform you that your manuscript has been deemed suitable for publication in PLOS ONE. Congratulations! Your manuscript is now with our production department. 

Kind regards, 

on behalf of

Dr. William J Brazelton 

Academic Editor

PLOS ONE